# The Symptom Structure of Postdisaster Major Depression: Convergence of Evidence from 11 Disaster Studies Using Consistent Methods

**DOI:** 10.3390/bs11010008

**Published:** 2021-01-13

**Authors:** Carol S. North, David Baron

**Affiliations:** 1The Altshuler Center for Education & Research, Metrocare Services, Dallas, TX 75247, USA; Carol.North@UTSouthwestern.edu; 2Department of Psychiatry, The University of Texas Southwestern Medical Center, Dallas, TX 75390-9070, USA; 3Department of Psychiatry, Western University of Health Sciences, Pomona, CA 91766, USA

**Keywords:** 9/11, postdisaster major depression

## Abstract

Agreement has not been achieved across symptom factor studies of major depressive disorder, and no studies have identified characteristic postdisaster depressive symptom structures. This study examined the symptom structure of major depression across two databases of 1181 survivors of 11 disasters studied using consistent research methods and full diagnostic assessment, addressing limitations of prior self-report symptom-scale studies. The sample included 808 directly-exposed survivors of 10 disasters assessed 1–6 months post disaster and 373 employees of 8 organizations affected by the September 11, 2001 terrorist attacks assessed nearly 3 years after the attacks. Consistent symptom patterns identifying postdisaster major depression were not found across the 2 databases, and database factor analyses suggested a cohesive grouping of depression symptoms. In conclusion, this study did not find symptom clusters identifying postdisaster major depression to guide the construction and validation of screeners for this disorder. A full diagnostic assessment for identification of postdisaster major depressive disorder remains necessary.

## 1. Introduction

Major depressive disorder (MDD) is an important public health problem [1], and is a main source of disability worldwide [1,2]. It is a major risk factor for suicide [3,4], and is associated with increased mortality worldwide [5]. Exposure to a disaster may contribute to MDD. MDD is the second most prevalent postdisaster disorder [6], found to occur in approximately 14–30% of disaster survivors in various studies [7,8]. Despite this strong association, posttraumatic stress disorder (PTSD) has received the most focus in disaster mental health research, and remains the most prevalent disorder reported after disasters [9,10].

A previous study examined statistical identifiers of major depression across various disasters, finding three main identifiers: (1) predisaster lifetime major depression, (2) disaster-related PTSD, and (3) indirect exposure through the disaster experience of a close friend or family member [6]. These identifiers of major depression following disaster exposure reflected psychosocial/interpersonal loss and bereavement and differ from identifiers of disaster-related PTSD that appear to represent the effects of direct personal disaster trauma exposure [6,7,8,11].

The phenotypic complexion of PTSD has received extensive investigation in studies of disasters and other types of trauma [12,13,14,15]. However, there is little agreement among clinical investigators about the symptom structure of PTSD across myriad factor analysis studies [16]. Less is known about the symptom structure of MDD in general [17], and even less is known about the symptom structure of MDD in the context of disasters. The United States National Institute of Mental Health has suggested that depression symptom groups could potentially identify symptoms that may be indicative of major mood disorders, subgroups of patients with distinctive illness characteristics, and responsiveness to various treatments [17]. Thus, identification of specific symptom patterns may have the potential to improve the detection of mood disorders, enable the selection of more effective interventions targeted for individual phenotypes [17], and provide important tools for genetic and pharmacological research [18]. In the context of disasters, the identification of individuals most likely to develop MDD can potentially guide mental health response efforts, which is well-appreciated [19,20]. Prior research has identified a number of external factors predicting MDD in disaster survivors [20,21,22,23,24], but the use of specific depressive symptoms to identify MDD has had little study and to date has provided little useful guidance [22]. Existing research has yielded incomplete or minimal agreement across MDD symptom factor studies [17,25]. In particular, work is needed to validate the results of symptom structure studies conducted using non-diagnostic symptom scales, with results from studies assessing full diagnostic symptom criteria [17]. No studies have investigated depressive symptom structures using structured diagnostic interviews in disaster-exposed samples.

The current analysis was conducted using a combined database from 11 previous disaster studies by this research team. These were a collection of studies designed to provide epidemiologic and descriptive findings pertaining to the prevalence and incidence of psychopathology and emotional distress in relation to exposure to disaster trauma, and this series of studies provided seminal knowledge informing the disaster mental health field about the postdisaster prevalence of different psychiatric disorders. However, these studies have not pursued investigation within the symptom structure of disaster-related MDD to examine subgroups or clusters of symptoms that might be potentially important identifiers of diagnosis or distinct illness phenotypes, or even clinical subgroups with distinct treatment responses. An earlier study of major depression examined external identifiers of the diagnosis of major depression using other variables in the data used for the current study [6]. The current study extends this work by examining the naturally-occurring symptom structure of major depression in disaster survivors, with the aim of using structural symptom patterns emerging from the analysis to guide mental health response efforts and assist with preliminary estimates of prevalence and incidence.

Specifically, the aims of this study were to: (1) characterize depression symptoms and symptom groups that statistically identify the postdisaster prevalence and incidence of major depression after disasters, (2) characterize clusters of symptoms that identify the likelihood of depressive illness following a disaster, and (3) compare the resulting symptom profiles from one database to another to determine the consistency of findings across different disaster samples and settings. The differentiation of postdisaster prevalence and incidence of major depressive symptoms and disorders in consideration of major depression symptom structures is of great relevance to disaster mental health research. This is because incident symptoms and disorders exclude much of the psychopathology that is likely to be unrelated to exposure to disaster trauma by virtue of an occurrence prior to the disaster.

## 2. Methods

The analyses for this study were conducted using 2 disaster databases with a full diagnostic assessment of 808 directly exposed survivors of 10 different disasters in one and 373 employees (with 27% directly exposed to disaster trauma, *n* = 163) recruited from 8 New York City organizations affected by the 9/11 terrorist attacks in the second. Combining and comparing data from these separate disasters was made possible by the use of largely consistent research methods applied across the individual disaster studies in the collection of the data by one research team. This methodological consistency generally achieved across these studies included: (1) timing of data collection as rapidly as feasible within the first few months after the disaster; (2) efforts to establish representative sampling methods including universal sampling, random sampling, and selection of participants from disaster-exposed households using governmental maps or lists of employees of affected businesses or official registries of disaster victims; and (3) use of structured interviews collecting systematic data on the same variables across all studies for disaster-related experiences and psychiatric disorders with onset and recency information keyed to the date of the disaster.

Extensive detail about the samples and the data collection methods is provided in previous publications [6,8,26]. Most importantly, all of these studies employed structured diagnostic interviews with onset and recency assessed relative to the date of the disaster, and structured interviews collecting disaster trauma exposures and other experience across the disasters. Where systematic sampling and timing of data collection were logistically not feasible, the closest methodological adjustments were made. The disasters in the 10-disaster database occurred between October 1987 and January 1994. Sampling for 6 of the 10 disaster sites in the first database was systematic, with a 77% participation rate. The 4 remaining samples in the first database, and the 9/11 sample in the second database, used convenience sampling with unknown participation rates. The 10 disaster incidents in the first database collectively represented the entire breadth of disaster typology (natural disasters, technological accidents, and intentionally human-caused disasters including mass shootings and terrorism); details about them are provided in a previous publication [26]. The 10-disaster sample was enrolled and interviewed 1–6 months after the disasters, but the 9/11 sample could not be interviewed until nearly 3 years after the disaster. Advance approval for the research was obtained from the Institutional Review Boards of the participating research institutions, and all members of the study sample provided written informed consent for participation.

Interviews of all participants were conducted using the major depression module of the Diagnostic Interview Schedule (DIS) assessing full diagnostic criteria of the established diagnostic criteria for major depression and its symptoms using *DSM-III-R* criteria [27] for the disasters that occurred before 1995 and *DSM-IV* criteria [28] for the disasters occurring in 1995 or later. The DIS Disaster Supplement [29] assessed details of the research participants’ disaster experience including disaster trauma exposures (direct, witnessed, indirect through close associates) according to PTSD diagnostic criteria. The DIS was able to differentiate predisaster vs. postdisaster depressive episodes because the onset and recency information were keyed to correspond to the date of the disaster. The original DIS used to collect data in the 10-disaster database queried only symptoms of the worst episode, and therefore postdisaster depressive symptoms in the 10-disaster database represent symptoms of only the identified worst lifetime depressive episode. For the calculation of the number of symptoms among the entire sample, those not describing a worst episode as occurring after the disaster were considered to not have postdisaster depressive symptoms. The 9/11 DIS was modified to allow a specific inquiry about symptoms of all postdisaster depressive episodes regardless of whether they represented the worst lifetime depressive episode.

### Data Analysis

Data analysis used SAS 9.4 (SAS Institute, Cary, NC, USA) software. The total combined sample included 1181 survivors of 11 disasters.

Multivariate logistic regression models nesting survivors within disasters (using PROC GLIMMIX in SAS 9.4 specifying a logit linkage function with binary distribution, nesting survivors within separate disasters in the 10-disaster database) were tested to identify postdisaster major depression (dependent variable) from the 9 criterion major depressive symptoms (of postdisaster episodes identified as the worst lifetime depressive episode for the 10-disaster database and for any postdisaster depressive episode for the 9/11 database) as independent covariates entered simultaneously into the models. Serial iterations of the models were tested by removing the least significant variable one by one until only significantly associated (*p* < 0.05) variables remained in the final model.

Exploratory factor analysis was conducted to identify factors associated with postdisaster major depression. For the 10-disaster database, factor analysis was conducted with major depressive symptoms of the worst episodes that occurred in the postdisaster period, and the factors were examined for their ability to identify postdisaster major depression. For the 9/11 database, factor analysis was conducted on the major depressive symptoms of all postdisaster depressive episodes, and the factors were examined for their ability to identify postdisaster major depression.

## 3. Results

### 3.1. Depressive Symptoms and Their Association with Postdisaster Major Depression

Table 1 and Table 2 show the prevalence of postdisaster depressive symptoms in the 10-disaster database and the 9/11 database, respectively. These tables provide findings for the entire sample and for subgroups without and with postdisaster major depression and with incident postdisaster major depression. Postdisaster major depression was diagnosed in nearly one third (30%) of the 9/11 sample but only 14% of the 10-disaster sample. In both databases, the majority of postdisaster major depression cases predated the disaster rather than arising as incident depression cases. Individuals without postdisaster major depression in either database had few postdisaster depressive symptoms. Among the individuals with postdisaster major depression in both databases, all symptoms except for thoughts of death/suicide among 9/11 survivors were endorsed by a majority of the survivors. Among survivors meeting the diagnostic criteria for a postdisaster major depressive episode, the average number of depressive symptoms was 7 in both databases.

Table 3 presents the results of multivariate models predicting postdisaster major depression as a dependent variable from individual depressive symptoms in the 10-disaster and 9/11 databases. For the 10-disaster database, the final multivariate logistic regression model predicting postdisaster major depression from the 9 *DSM-IV* symptoms of postdisaster episodes identified as the worst lifetime depressive episode yielded psychomotor agitation/retardation and suicidal ideation as significant predictors of postdisaster major depression. Application of the model to the identification of incident major depression specifically yielded only psychomotor agitation/retardation as significantly associated.

In a similar analysis of postdisaster depressive symptoms in the 9/11 database, the final multivariate logistic regression model predicting postdisaster major depression as a dependent variable from the 9 *DSM-IV* symptoms of any postdisaster depressive episode yielded loss of interest or pleasure, appetite disturbance, sleep disturbance, and fatigue as significant predictors of postdisaster major depression. Application of this model to the identification of incident major depression specifically yielded loss of only interest or pleasure and appetite disturbance as significantly associated.

### 3.2. Exploratory Factor Analysis to Chracterize Factors Identifying Postdisaster Major Depression

Exploratory factor analysis in the 10-disaster database was conducted on the 9 major depressive symptoms of postdisaster episodes identified as the worst lifetime depressive episode (not shown in the tables). In this analysis, the first factor explained 73% of the variance among the depressive symptom items. This factor loaded relatively equally on all 9 symptom items (scoring coefficients’ absolute values ranging from 0.11 to 0.14) and its mean factor scores were positively associated with postdisaster major depression (t = 7.20, df = 114, *p* < 0.001).

Exploratory factor analysis was also conducted on the 9 postdisaster major depressive symptoms in the 9/11 database. The first 3 factors in this database explained 88% of the variance in the symptom data. Only the first factor, which loaded relatively equally on all 9 symptom items (scoring coefficients ranging from 0.08 to 0.13), had a mean factor score that was significantly (positively) associated with postdisaster major depression (t = 27.04, df = 117.42, *p* < 0.001).

Thus, neither database provided meaningful or useful subgroups or clusters of postdisaster depressive symptoms within the diagnostic criteria for major depression for the identification of the postdisaster prevalence or incidence of this disorder.

## 4. Discussion

This study examined the symptom structure of major depression across two databases representing a total of 1181 survivors from 11 disasters studied using consistent research methods with the collection of full diagnostic data. Despite the much higher proportion with direct trauma exposure in the 10-disaster database (100%) compared to the 9/11 database (44%), the postdisaster prevalence of major depression was much higher (30%) in 9/11 disaster survivors than in survivors in the 10-disater database (14%). Less than one half of the postdisaster major depression represented incident depressive disorders arising after the disaster, indicating that major depression was largely not a product of the disaster. Because the numbers of most postdisaster depressive symptoms in both databases among survivors who did not have major depression were essentially equivalent and relatively small (0.1 symptoms on average), most depressive symptoms largely reflected the psychopathology of major depression rather than nonpathological emotional distress.

In the 10-disaster database, psychomotor agitation/retardation and suicidal ideation were independently associated with postdisaster major depression, but only suicidal ideation was independently associated with incident postdisaster major depression. Entirely different variables were found to be independently associated with major depression in the 9/11 database, including loss of interest or pleasure (anhedonia), appetite disturbance, sleep disturbance, and fatigue, but only loss of interest or pleasure (anhedonia) and appetite disturbance were independently associated with incident postdisaster major depression. Because incident symptoms are those most likely to be disaster-related, the findings suggest that suicidal ideation, anhedonia, and appetite disturbance may be of particular interest for the identification of MDD, but replication by other studies is needed. The findings of the current analyses are somewhat consistent with the results of a latent class analysis of data from self-reported depressive symptom surveys of Chinese undergraduates (not a disaster study) that found depressed mood, anhedonia, and suicidal ideation to be associated with more severe depression [30]. Factor analysis in both databases of the current study, however, showed that the factors that emerged loaded relatively equally on all 9 symptom items, not identifying any specific clusters of depression symptoms that could be of potential use for identification of MDD.

A strength of this study was its nosological approach to studying major depression, querying symptoms only within depressive episodes established using full diagnostic criteria. Other studies have investigated the ability of self-report depression screening tools to identify MDD among disaster and trauma survivors with conflicting results, and cautions have been raised about the use of these measures for psychiatric diagnosis or making clinical decisions [19,20,31]. The methodological strength of using full diagnostic structured interviews to examine the symptom structure of major depression is unique to this study. A related strength of this study was the differentiation of postdisaster prevalence from postdisaster incidence, the latter of which excludes much of the psychopathology that is likely to be unrelated to exposure to disaster trauma by virtue of its occurrence prior to the disaster. This is an important nosological consideration that has been widely unaddressed in prior disaster research.

The size of this combined disaster database providing a full diagnostic assessment in this study is large given the resource burden required for the collection of full diagnostic data. Most studies of this size do not have the extensive time and personnel resources to devote to conducting structured diagnostic interviews and instead substitute lower-cost self-report symptom scales that do not provide diagnosis. The cost of conducting diagnostic interviews is generally prohibitive for single studies of this size, a limitation overcome by combining diagnostic data from numerous smaller studies for this analysis. Both the combination of survivor samples of 10 different disasters into one database and a replication of the analysis in a second disaster database added variability to the representation of survivors and their characteristics across disasters of different types and locations, and permitted a comparison of the findings in a second disaster database. However, a more definitive investigation of the specific effects of these different characteristics on postdisaster depressive symptom structures must await further studies with analyses focused on this issue.

Although both *DSM-III-R* and *DSM-IV* criteria were used for collection of the data in the different disaster studies, the criteria sets use identical definitions of the depressive episode and the same 9 criterion symptoms. All of the data were collected before the establishment of *DSM-5* criteria; thus, further study is needed to test the symptom structure of depressive symptoms using current criteria, although few differences might be expected because of the relative stability of diagnostic criteria for major depression. The inherent construction of the DIS limited the analysis of the 10-disaster database because only the worst lifetime episodes were queried, but a modification of the interview for the 9/11 study allowed query of postdisaster major depressive episodes regardless of whether they represented the worst episode. This allowed more specific examination of the time frame of interest in the 9/11 study, but it may have introduced differences in the results found between the 10-disaster and the 9/11 databases.

Importantly, it should be noted that the retrospective nature of the collection of predisaster and postdisaster symptoms at the time of the interviews does not strictly permit “prediction” of future diagnosis in these databases, but any associations found (none in this study, however) could potentially provide the basis for the identification of current diagnoses from current subgroups or clusters of symptoms reported, or even inspire future research on the prospective ability of symptom groups to predict future diagnoses. There is always a limitation of potential recall bias in interviews of participants about themselves, especially with a greater amount of time elapsed between the disaster and the collection of data. There was inevitable variation in the timing of the interviews with respect to the number of months between the date of the disaster and collection of the data, necessitated by logistical barriers inherent in specific disaster settings, and the 9/11 disaster database was not collected until almost 3 years after the incident, which may have introduced recall bias in this database not present in the 10-disaster database. Although some of the disaster samples had systematic recruitment and enrollment of participants with high participation rates, some constituted volunteer samples with low or unknown participation rates, which may have introduced selection biases. The two databases differed in disaster trauma exposure levels, with direct exposures for all participants of the 10-disaster database but only 44% of the 9/11 database, potentially contributing further to differences in findings across these databases.

The identification of population groups at risk for postdisaster MDD based on characterization by presenting symptom clusters could potentially lead to significant public health advances through the construction and validation of postdisaster MDD screening tools. This has been successfully achieved for PTSD in a prospective study of PTSD symptoms and diagnosis [32], and this type of investigation in postdisaster MDD might also be expected to provide advances in both the identification and prediction of disaster-related MDD cases with similar public mental health benefits for these populations. Focusing limited mental health resources on high-risk groups identified by such tools would be more efficient than applying resource-intensive mental health interventions across entire populations. Unfortunately, the current study did not yield consistent patterns of symptoms identifying postdisaster major depression, and both database factor analyses found that depression symptoms grouped cohesively without the clustering of symptom subgroups that might potentially identify disaster-related MDD. Further, no other disaster mental health research to date has consistently identified symptom clusters that coalesce into distinct factors or consistently identify major depression. Given the lack of progress toward these goals to date, a full diagnostic assessment of MDD after disasters remains necessary to identify cases, because symptom-based shortcuts have not been found [19,22].

It is unclear why this and prior studies have not yet been able to identify symptom groups or clusters that may signal the likelihood of postdisaster MDD. It may be that additional studies of a variety of disasters such as this one using full diagnostic assessment are still needed, but with attention to address additional methodological issues, such as the need for an assessment of all postdisaster episodes rather than the worst episodes occurring in the postdisaster setting, consistently earlier timing of postdisaster psychiatric assessment, and comparable disaster trauma exposure across samples. It is also possible that symptom cluster solutions will not be able to provide more ready identification of MDD, so that further methodological refinements of future studies will fail to demonstrate the achievement of this desired goal. It is possible to imagine that the development of futuristic biological detection technologies may ultimately be needed for more efficient detection of postdisaster MDD.

## 5. Conclusions

This study overcame important methodological limitations of prior studies that have relied on symptom screeners, but still did not successfully identify symptom clusters or consistent symptoms identifying the likelihood of postdisaster major depression to guide the construction and validation of symptom screening tools for the identification of MDD in disaster-exposed populations. At this point, because no symptom subset has been identified for a more efficient identification of the likelihood of postdisaster MDD, full diagnostic assessment of postdisaster MDD remains necessary for the identification of this important disorder in the context of disaster settings.

## Figures and Tables

**Table 1 behavsci-11-00008-t001:** 10-disaster database prevalence of postdisaster depressive symptoms (in worst episode only).

Depressive Symptoms in the Worst Episodes that Occurred Post Disaster	Full Sample *N* = 808	No Postdisaster Major Depression *N* = 695 (86%)	Postdisaster Major Depression *N* = 113 ^a^ (14%)	Incident Major Depression *N* = 45 ^b^ (6%)
% (*n*)				
Depressed mood	7 (57)	3 (18)	98 (39)	90 (9)
Loss of interest/pleasure	4 (29)	1 (7)	55 (22)	60 (6)
Appetite disturbance	5 (41)	1 (10)	78 (31)	90 (9)
Sleep disturbance	6 (50)	2 (13)	93 (37)	80 (8)
Psychomotor agitation/retardation	3 (28)	1 (4)	60 (24)	90 (9)
Fatigue/loss of energy	5 (43)	1 (10)	83 (33)	90 (9)
Guilty/worthless feelings	3 (27)	1 (5)	55 (22)	60 (6)
Poor concentration	6 (50)	2 (13)	93 (37)	90 (9)
Thoughts of death/suicide	4 (36)	1 (5)	78 (31)	60 (6)
Mean (SD)				
Number of depressive symptoms	0.4 (1.7)	0.1 (.8)	6.9 (1.3)	7.1 (1.4)

^a^ Only 40 were assessed for specific symptoms. ^b^ Only 10 were assessed for specific symptoms.

**Table 2 behavsci-11-00008-t002:** 9/11 database prevalence of postdisaster depressive symptoms.

Depressive Symptoms in Postdisaster Episodes	Full Sample *N* = 373	No Postdisaster Major Depression *N* = 262 (70%)	Postdisaster Major Depression *N* = 111 (30%)	Incident Major Depression *N* = 38 (10%)
% (*n*)				
Depressed mood	26 (96)	3 (8)	79 (88)	87 (33)
Loss of interest/pleasure	24 (90)	2 (6)	76 (84)	89 (34)
Appetite disturbance	25 (95)	1 (3)	79 (88)	92 (35)
Sleep disturbance	27 (102)	1 (3)	85 (94)	92 (35)
Psychomotor agitation/retardation	24 (92)	2 (5)	77 (85)	89 (34)
Fatigue/loss of energy	26 (99)	4 (4)	81 (90)	82 (31)
Guilty/worthless feelings	18 (69)	0 (0)	60 (67)	79 (30)
Poor concentration	28 (105)	3 (7)	84 (93)	92 (35)
Thoughts of death/suicide	8 (31)	0 (0)	27 (30)	29 (11)
Mean (SD)				
Number of depressive symptoms	2.0 (3.2)	0.1 (0.7)	6.5 (2.4)	7.5 (1.2)

**Table 3 behavsci-11-00008-t003:** Multivariate models predicting postdisaster and incident major depression in the 10-disaster and the 9/11 databases ^a^.

	β	SE	t	df	*p*	OR	95% CL
***10-disaster database*^b^**							
Postdisaster major depression							
Psychomotor agitation/retardation	2.85	0.64	4.49	796	<0.001	17.33	4.98, 60.31
Suicidal ideation	3.27	0.54	6.06	796	<0.001	26.25	9.10, 75.67
Incident major depression							
Psychomotor agitation/retardation	2.07	0.46	4.48	796	<0.001	7.95	3.21, 19.69
***9/11 database*^c^**							
Postdisaster major depression							
Loss of interest or pleasure	1.66	0.78	2.12	368	0.035	5.27	1.13, 24.65
Appetite disturbance	2.71	0.86	3.13	368	0.002	15.02	2.76, 81.85
Sleep disturbance	2.60	0.83	3.14	368	0.002	13.49	2.65, 68.79
Fatigue	2.85	0.77	3.72	368	<0.001	17.34	3.83, 78.50
Incident major depression							
Loss of interest or pleasure	4.10	1.14	3.59	370	<0.001	60.05	6.38, 565.30
Appetite disturbance	1.60	0.73	2.20	370	0.028	4.96	1.19, 20.69

^a^ Only variables significantly associated with major depression are listed in the table; ^b^ Symptoms of the worst lifetime episode; ^c^ Symptoms of any postdisaster episode.

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
