# Peer review of "The Symptom Structure of Postdisaster Major Depression: Convergence of Evidence from 11 Disaster Studies Using Consistent Methods"

_behavsci, 2021, doi:10.3390/bs11010008_

Round 1
Reviewer 1 Report
The Manuscript „The symptom structure of postdisaster major depression: Convergence of evidence from 11 disaster studies using consistent methods” deals with symptoms and predictive value of factors for major depressive disorders after disasters.
I believe the Manuscript deals with an interesting topic, but should be re-written using clearer and to-the-point language. In addition, the Authors should explain how this study differs from previous studies published regarding the same topic and using the same databases. The Methods should be clear, especially regarding the time of the exposure, symptoms, and diagnoses – if the Authors are claiming a causal relationship.
General comments:
Please make sure the citation style follows Journal guidelines.
The Author(s) have clearly dealt with the topic at hand previously, and have cited their own work extensively in the Introduction and Methods. Therefore, it should be stated more clearly what the previous work was about, what were the previous findings, what were the limitations or further investigations necessary, how this work is different from the previous work, and what this work adds. This is especially important considering that the databases used seem the same as almost 10 years ago.
Introduction
Although the Authors have provided interesting information regarding MDD and PTSD in the Introduction, the text just does not seem to flow, the phrases are complicated and the information is not transmitted to the reader correctly.
E.g. Identification of specific symptom patterns may improve detection of mood disorders. In addition, it could enable selection of more effective interventions targeted for individual phenotypes (Shafer, 2006), and provide important tools for genetic and pharmacological research (Korszun et al, 2004).
The Authors should clearly state the Aim of their study as the last paragraph of the Introduction. Currently, the last paragraph of the Introduction should be incorporated into the Methods section.
The Authors should not state the “goal” but clear aim, and should be careful if they have “done what they promise”.
Methods
The methods are well described, but it would be nice to hear a simple explanation regarding the “consistent methods”, as this is mentioned throughout the text. What goes under this consistent methodology which is used?
Why are Tables 1 and 2 already cited in the Methods? They are again mentioned at the beginning of the Results.
I am unsure of the study design used in the various databases. Were the participants enrolled and interviewed right after the disaster, and then followed to see if they develop MDD or PTSD? Or were they enrolled some time after, diagnosed, and asked about the symptoms?
Are the symptoms those occurring right after the disaster, or at the time of the interview, or when?
Results
Tables could be made to be more appealing. They seem like the raw output of SAS.
I am unsure what was the conclusion of the Authors after the factor analysis regarding the factors predicting postdisaster major depression... Is there some table or figure showing the data of the factor analysis?
Discussion
The Authors discuss the proportion of participants with direct trauma, but could this be due to sampling? They did not include ALL survivors of various disaster to be able to compare their exposure with the outcome?
I find it still unclear when did the exposure occur, when did the symptoms occur, and when were the diagnoses made. These time determinants should be state clearly throughout the Manuscript.
Explaining better the study design is also necessary to make sure we can discuss “prediction” of major depression, as compared to associations with major depression.
There are still various unclear phrases throughout the text. E.g. The size of this combined disaster dataset providing full diagnostic assessment is large given the resource burden required for collection of full diagnostic data. The nesting of the survivors within 10 different disasters in the analysis and comparison from one disaster database to another allowed the analysis to account for variability across disasters.
It is difficult to understand what the last paragraph of the Discussion means. The Authors have tried, but have failed, but their attempt will help them in the future? How? Why?
Conclusions
I am unsure how the conclusions arise from the results – what was overcome and how?
What does it mean: “At this point, full diagnostic assessment of postdisaster major depressive disorder remains necessary for the identification of this important disorder in the context of disaster settings”? As opposed to what?
Author Response
Please, see the attachment.

Reviewer 2 Report
It has been said that the structure of disaster mental health research makes it difficult to create EBM. This study is a meaningful attempt to improve that. The study is highly regarded for its use in understanding the structure of depression during a disaster.
Thank you for submitting your excellent paper.
83-86
Interviews of all participants were conducted using the major depression module of the Diagnostic Interview Schedule (DIS) assessing DSM-III-R criteria(Robins et al., 1998) for the disasters that occurred before 1995 and DSM-IV criteria (Robins et al., 2000) for the disasters occurring in 1995 or later.
→
All participants were interviewed using the Major Depression module of the Diagnostic Interview Schedule (DIS), which assesses the DSM-III-R criteria for disaster (Robins et al., 1998). Although the reason it was III-R rather than DSM-4-R or DSM-5 was that these new diagnostic criteria were not yet available at the time of data collection, is it possible that the new diagnostic criteria could be evaluated in the future? If so, is the current data that came out in III-R strongly influenced by that and could it change the interpretation?
What are the ten disasters?
Is it a natural disaster, a man-made disaster, or a complex one?
I hope you can clarify the difference between the two.
Wouldn't that change the frequency of PTSD?
Doesn't it also mean that the frequency of subsequent depression will be different?
Can you think of any such differences between 9/11 and 10 disasters?
Depression is also associated with a history of alcohol and drugs in addition to PTSD, but is there a relationship between the two? What can you learn from basic patient data?
120-123
Postdisaster major depression was diagnosed in nearly one third (30%) of the 9/11 sample but only 14% of the 10-disaster sample. In both databases, the majority of postdisaster major
depression cases predated the disaster rather than arising as incident depression cases. Individuals without postdisaster major depression in either database had few postdisaster depressive symptoms.
→
Is there any reason for this difference in the data set, number of deaths, regional differences, sex ratio, age distribution, or underlying disease?
It is also important to consider when the survey was conducted after the disaster.
After some time, a number of different life contextual influences may be involved.
Is it possible that major depression reflects the original community depression? T
here may also be a higher prevalence of adjustment disorders with depression, that are not major depression, but this may be a limitation of the study.
229 -232
This study overcame limitations of prior studies using symptom screeners, but it did not identify symptom clusters or consistent symptoms predicting postdisaster major depression to guide construction and validation of symptom screening tools for identification of major depressive disorder in disaster-exposed populations.
→
Thank you for this very important challenge.
Do you have any other recommendations for future research on disasters and depression as a result of this study?
Author Response
Please, see the attachment.

Reviewer 3 Report
This is an interesting study of symptoms of post disaster major depressive disorders (MDD). The study is the extension of their previous work published in the Journal of Psychiatric Research in 2018. The manuscript should be improved prior considering for publication, in particular, the rationale od the study in the introduction section and discussion of the findings with the relevant literature, as well as to restructure and add necessary information in the methods.
- The first statement in the abstract is strong, but it is not fully explored in the main document.
- Please, explain the problem with the symptom structure of MDD. Why this structure is the problem? Are the symptoms and predictors of MDD so different in the literature?
- Can you say more about the conceptual and practical importance of distinguishing the symptom structure of predisaster MDD from the symptom structures of disaster-related MDDs, indirect exposure MDDs, and post-disaster MDDs?
- What could be the value of the existence of harmonized symptom structures for MDD? Is this possible at all given that we live in world of inequalities?
- Line 44: Please, add country of the National Institute of Mental Health.
- Regarding ethical aspects, it would be important to clearly distinguish the value of this study in contrast to the previous one which had the same study sample (https://doi.org/10.1016/j.jpsychires.2017.12.013)
- Highlight the novelty of this research and please, make sure the manuscript does not use the same sentences as in https://doi.org/10.1016/j.jpsychires.2017.12.013
- Line 70: Please, structure the Methods section paragraphs, for instance:
- study design and type of study
- sampling method (inclusion-exclusion factors)
- study sample and variables
- study instrument
- data sources
- data analysis
- Consistent methods are stressed in the title. Can you say more about approaches used in analysing structure symptoms of MDD? Please, explain 'consistent methods' applied in this study.
- Lines 86-87: I do not understand, what did you want to say with these two sentences? Tell readers why the supplement is relevant for this study. Also, explain why you were using PTSD criteria, and identify the criteria.
- Lines 89-95: Please, this is important to be mentioned again in the discussion section. State the related biases and study limitations.
- Please, expand on the study limitations.
- Discussion is week. There is no discussion of findings with similar studies.
- Practical and political implications of the study findings are missing.
- In the conclusion, you mentioned previous studies, but there was not a single reference in the discussion section.
- The literature is old. Some great papers published in recent years are not cited.
Author Response
Please, see the attachment.

Round 2
Reviewer 1 Report
The Authors have adequately responded to my comments and have made significant improvements to the Manuscript.
Author Response
Please, see the attachment.

Reviewer 3 Report
The authors have put efforts to improve manuscript clarity and value. This was a challenging task, as they used data from a combined disaster database, there is a significant time lag between data collection and the writing of this study, and their study design, despite their aim ("to identify symptom grouping or clustering of potential utility for efficiently identifying the likelihood of major depression.") "did not find symptom clusters identifying postdisaster major depression to guide construction and validation of screeners for this disorder. Full diagnostic assessment for identification of postdisaster major depressive disorder remains necessary". Nonetheless, the paper is worth reading, as it nicely describes various efforts so far put in this field and points out directions to future research not only in clinical medicine but also in public health.
I suggest rephrasing the aim of the study in order to better reflect the study value. For instance, "This study examined the existence and utility of MDD symptom structures in disaster survivors with the hopes of finding structural information in symptom patterns that could possibly guide mental health response efforts." (that was one of the replies to my comments). In addition, it identifies the postdisaster prevalence and incidence of major depression after disasters.
Next, for the readability, it is better to avoid long sentences such as the one from line 78 to line 82.
I could not find the years of the conducted interviews with 808 directly-exposed survivors of 10 different disasters.
Author Response
Please, see the attachment.
